

# A short, robust brain activation control task optimised for pharmacological fMRI studies

Jessica-Lily Harvey[1,2], Lysia Demetriou[3,4], John McGonigle[2,5] and Matthew B. Wall[2,3,6]

[1] School of Psychology and Neuroscience, University of St. Andrews, St Andrews, United Kingdom
[2] Division of Brain Sciences, Imperial College London, London, United Kingdom
[3] Invicro Ltd., London, United Kingdom
[4] Department of Medicine, Imperial College London, London, United Kingdom
[5] Perspectum Diagnostics, Oxford, United Kingdom
[6] Clinical Psychopharmacology Unit, University College London, University of London, London, United Kingdom

Corresponding author
Matthew B. Wall,
matthew.wall@imperial.ac.uk

## ABSTRACT

**Background**. Functional magnetic resonance imaging (fMRI) is a popular method for examining pharmacological effects on the brain; however, the BOLD response is dependent on intact neurovascular coupling, and potentially modulated by a number of physiological factors. Pharmacological fMRI is therefore vulnerable to confounding effects of pharmacological probes on general physiology or neurovascular coupling. Controlling for such non-specific effects in pharmacological fMRI studies is therefore an important consideration, and there is an additional need for well-validated fMRI task paradigms that could be used to control for such effects, or for general testing purposes.

**Methods**. We have developed two variants of a standardized control task that are short (5 minutes duration) simple (for both the subject and experimenter), widely applicable, and yield a number of readouts in a spatially diverse set of brain networks. The tasks consist of four functionally discrete three-second trial types (plus additional null trials) and contain visual, auditory, motor and cognitive (eye-movements, and working memory tasks in the two task variants) stimuli. Performance of the tasks was assessed in a group of 15 subjects scanned on two separate occasions, with test-retest reliability explicitly assessed using intra-class correlation coefficients.

**Results**. Both tasks produced robust patterns of brain activation in the expected brain regions, and region of interest-derived reliability coefficients for the tasks were generally high, with four out of eight task conditions rated as 'excellent' or 'good', and only one out of eight rated as 'poor'. Median values in the voxel-wise reliability measures were also > 0.7 for all task conditions, and therefore classed as 'excellent' or 'good'. The spatial concordance between the most highly activated voxels and those with the highest reliability coefficients was greater for the sensory (auditory, visual) conditions than the other (motor, cognitive) conditions.

**Discussion**. Either of the two task variants would be suitable for use as a control task in future pharmacological fMRI studies or for any other investigation where a short, reliable, basic task paradigm is required. Stimulus code is available online for re-use by the scientific community.

## INTRODUCTION

Functional Magnetic Resonance Imaging (fMRI) is currently one of the major standard methods in cognitive neuroscience research. FMRI provides reasonably high spatial and temporal resolution data, is flexible enough to accommodate a wide variety of experimental designs, and exposure to magnetic fields presents no danger to most subjects (*Logothetis, 2008*; *Soares et al., 2016*). FMRI can also be used as an index of pharmacological effects; drugs or hormones can be administered before or during a scanning session, and the results compared with a baseline or placebo session (e.g., *Carhart-Harris et al., 2014*; *Comninos et al., 2017*; *Kaelen et al., 2016*; *Upadhyay et al., 2011*). Pharmacological-fMRI studies may be used in the drug discovery process (*Wise & Tracey, 2006*; *Matthews, Rabiner & Gunn, 2011*; *Carmichael et al., 2018*), in the characterization of the effects of commonly-prescribed drugs (*Maron et al., 2016*), or in the exploration of disorders such as addiction (*Quelch et al., 2017*).

Conducting pharmacological-fMRI investigations presents many of the same challenges as standard fMRI, but also has some unique issues. One fundamental concern is related to the fact that (most commonly) fMRI studies use the BOLD (Blood-Oxygen-Level-Dependent) signal as the primary end-point. This is a contrast produced by local changes in the ratio of oxygenated and de-oxygenated hemoglobin (*Buxton, Wong & Frank, 1998*; *Friston et al., 2000*), and is usually regarded as a proxy measure of neural activity. However, the relationship between neural activity and this vascular response (neurovascular coupling) is complex and relies on a number of cellular and metabolic processes (*Logothetis et al., 2001*). Use of a pharmacological agent combined with fMRI means that any differences observed in the BOLD response may be a combination of direct neural effects of the drug (usually the effects of interest), and indirect effects of the drug (e.g., on neurovascular coupling, or global, systemic effects on blood-pressure, cerebral blood flow, heart-rate, etc.; usually regarded as confounding effects). One example is caffeine, which has direct neural effects on adenosine $A_1$ and $A_{2a}$ receptors, but is also a powerful cerebral vasoconstrictor (*Diukova et al., 2012*). Separating the neural and vascular effects of even such a well-studied drug as caffeine is therefore a considerable challenge. For detailed reviews of these issues see *Bourke & Wall (2015)*, and *Iannetti & Wise (2007)*.

One potential approach to this problem is to simply use an alternative imaging method. Recent work by *Stewart et al. (2014)* has shown that Arterial Spin Labelling (ASL) MRI may be a superior technique to assess some pharmacological effects, largely because of its stability across longer periods of time. Other techniques such as magnetoencephalography (MEG) show great promise (*Baillet, 2017*) but have yet to achieve the broad availability and application of BOLD-fMRI. The flexibility of BOLD-fMRI in terms of task paradigms, its 'good-enough' spatial and temporal resolution (*Logothetis, 2008*), and its broad availability and relative ease of use mean it is still a popular method in pharmacological investigations.

While BOLD-fMRI experiments typically incorporate intrinsic control conditions, these may not adequately control for some hypothetical pharmacological effects. If the drug produces a simple amplitude shift in the responses to both (experimental, and control) conditions, the control is still effective and the effect on the experimental >control contrast will be negligible. However, if the drug (a) selectively affects one condition, and not the other, or (b) produces a non-linear effect on the difference between the conditions, the intrinsic control may become unreliable. Also, many experiments have used a simple baseline (resting) control condition (e.g., *Wise, Williams & Tracey, 2004*). In such cases, amplitude shifts to the baseline as a result of non-neural effects could be very problematic. One method of mitigating this problem is the use of an independent control task paradigm as part of a pharmacological fMRI scanning session (*Iannetti & Wise, 2007*). For example, *Murphy et al. (2009)* used a visual control task in their study of the effect of citalopram on amygdala responses to emotional faces. In this case the lack of effect of the drug on the visual control task suggests that the effects seen in the main task are unlikely to be due to effects on neurovascular coupling, or other global/systemic effects. However, the use of a single (visual) control task, which gives activation in a circumscribed region of the brain (the occipital lobe) is suboptimal as effects on neurovascular coupling may conceivably vary across the brain. *Comninos et al. (2017)* used a much more elaborate control task (based on *Pinel, Thirion & Meriaux, 2007*) in their recent study on the sex hormone kisspeptin. This task involved ten trial conditions which gave results in five separate functional domains (visual, auditory, language, motor, and cognitive), and in a much wider spatial distribution across the brain. This task involved relatively complex instructions for the subjects, and also included some culturally-specific language stimuli, which somewhat limits its broad applicability.

An ideal task for the control of pharmacological fMRI studies should have the following characteristics. First, it should be short in duration as it generally has to be included as part of a broader set of functional task paradigms, anatomical scans, and perhaps other MRI measures (resting-state fMRI, perfusion measures, spectroscopy etc.). Second, it should be simple, both for the subject to perform and for the experimenter to run and analyse. It should require no complex instructions and depend upon only standard equipment (non-specialist computer hardware/software, audiovisual systems, and simple response devices). Third, it should contain a number of different trial types, which produce activation in different brain networks, in as wide a spatial distribution across the brain as possible. This helps to rule out effects on neurovascular coupling which may differ in spatially remote brain regions. Fourth, it should be general-purpose; applicable to a wide range of different pharmacological (or other) fMRI studies. Fifth, it should be reliable; it should produce robust results within a single-session, and produce reliable results across multiple sessions. This last point is of particular importance, as use of an unreliable control task would constitute an additional confound, however no previous pharmacological fMRI study has explicitly assessed the reliability of its control task. Indeed reliability is relatively seldom formally assessed in fMRI studies (*Plichta et al., 2012*).

Our aim was to develop some basic open-source task paradigms that have been formally evaluated in terms of reliability, for future use in pharmacological studies, or

any investigation where simple test paradigms are required, and reliability is an important consideration (e.g., multisite fMRI studies; *Brown et al., 2011*). We have developed two variants of a task paradigm that meet the above mentioned criteria, and are furthermore programmed in an open-source software environment (PsychoPy; *Peirce, 2007*; *Peirce, 2008*). One variant consists of visual, auditory, motor, and eye-movement trials. The other substitutes a brief working-memory task for the eye-movement trials, but is otherwise identical. Both are short (5 minutes in duration), simple (requiring only standard audiovisual equipment, and a single-button response box), and both produce four robust, distinct, and specific patterns of brain activation in widely-distributed brain regions. The reliability of the task variants across two scanning sessions has been explicitly assessed using a combination of voxel-wise and Region of Interest (ROI) based approaches.

## METHODS

### Subjects

Fifteen healthy subjects (six males, nine females) from ages 21–48 (mean age = 30) were scanned on two separate occasions with the average re-test interval being two weeks. All participants were fully briefed and provided written informed consent. All scans were performed under local institution-approved guidelines for MRI scans on healthy subjects to ensure adherence to ICH-GCP standards.

### Task design and procedure

The tasks were programmed in PsychoPy (*Peirce, 2007*; *Peirce, 2008*); a free, open-source, cross-platform Python library optimized for experimental design. The task was a randomized event-related design, and consisted of 5 discrete trial types: auditory, visual, motor, cognitive and null trials, each lasting exactly three seconds. A small red, square fixation point was present throughout each task (except in one trial type, as noted below) at the centre of the screen. Auditory trials presented six pure tones for 0.5s each, at frequencies of 261.63 Hz, 293.66 Hz, 329.63 Hz, 349.23 Hz, 440 Hz, and 493.88 Hz (corresponding to the musical pitches $C_4$, $D_4$, $E_4$, $F_4$, $A_4$, and $B_4$, respectively). The order of the six tones was randomly determined on each trial. Visual trials consisted of a centrally-presented sine-wave grating subtending approximately 10° of visual angle and with a spatial frequency of 1.2 cycles/degree. The grating drifted laterally at a rate of 6 cycles per second, and the direction of drift reversed every 0.5s. Motor trials consisted of three presentations of a small image of a button, presented just above the centre of the screen, for 1s each. This was a cue for subjects to press the response box key, and the button image disappeared after each response was made. The 'cognitive' trial differed in the two variations of the task. In the eye-movement variant, the fixation point moved to six different locations corresponding to the compass locations North–East, East, South–East, North–West, West, and South–West. These points were mapped on a circle with a radius of approximately 8.75° of visual angle. Each location was maintained for 0.5 s, and all six were presented (in a random order) in each three second trial. In the working-memory variant of the experiment, the cognitive trial consisted of a brief working memory task. This involved the presentation of two letter strings (containing four letters each), followed by a single letter. The subject's task was to

indicate whether the final, single letter was present in the first letter string. If the final letter was present in the first letter string, they were instructed to push the response button. If the final letter was not present in the first letter string they were instructed to make no response. For half the working memory trials the final letter was present in the first string, and for half it was not present. Finally, in the null trials the fixation point was maintained for three seconds, with no other stimuli presented.

The two task variants were identical, except for the inclusion of eye-movement trials in one, and working-memory trials in the other. Each task consisted of 100 trials (20 of each of the four active conditions, plus 20 null trials) presented in a standardized pseudo-random order. Separate versions of the two tasks reversed the trial order, and the order of presentation of these versions was counter-balanced across subjects and scans. The order of presentation of the two task variants in the scan sessions was also systematically varied across subjects and scans. The task durations were exactly five minutes (100 trials of 3 s duration) plus a 10 s buffer period at the end.

Prior to each scan session, subjects were shown a short demonstration version of each variant of the task, and instructed on how to perform them. Written instructions were also presented to the subjects in the scanner, immediately prior to the start of each task (see Supplemental Information for details). During the scanning session, visual stimuli were projected through a wave guide in the rear wall of the scanner room onto a screen mounted in the rear of the scanner bore. This was viewed in a mirror mounted to the head coil. Participants received auditory stimuli via MRI-compatible headphones, and responded using a one-button response box held in their right hand. Responses were recorded using PsychoPy's data-logging routines.

All the tasks' PsychoPy code and associated files are available on Figshare at Wall, Matthew (2017): fMRI_control_task.zip. figshare. Code. https://doi.org/10.6084/m9.figshare.5162065.v1.

It is also availble on GitHub: https://github.com/mattwall1103/fMRI-Control-Task.

## MRI data acquisition and analysis

Data were acquired on a Siemens 3T Magnetom Trio MRI scanner (Siemens Healthcare, Erlangen, Germany), equipped with a 32-channel phased-array head coil. A high-resolution T1-weighted image was acquired at the beginning of each scan using a magnetization prepared rapid gradient echo (MPRAGE) sequence with parameters from the Alzheimer's Disease Research Network (ADNI; 160 slices $\times 240 \times 256$, TR $= 2,300$ ms, TE $= 2.98$ ms, flip angle $= 9°$, 1 mm isotropic voxels, bandwidth $= 240$ Hz/pixel, parallel imaging factor $= 2$; *Jack et al., 2008*). Functional data collection used an echo-planar imaging (EPI) sequence for BOLD contrast with 36 axial slices, aligned with the AC-PC axis (TR $= 2,000$ ms, TE $= 31$ms, flip angle $= 80°$, 3 mm isotropic voxels, parallel imaging factor $= 2$, bandwidth $= 2,298$ Hz/pixel). Each functional scan lasted five minutes and ten seconds and consisted of 155 volumes.

Analysis was completed with FSL version 5.0.4 (FMRIB's software Library; Oxford Centre for Functional Magnetic Resonance Imaging of the Brain; http://www.fmrib.ox.ac.uk/fsl/). Anatomical Images were initially skull-stripped using BET (Brain Extraction Tool; included

in FSL). Images were pre-processed with standard parameters (head-motion correction, 100 s temporal filtering, 6 mm spatial smoothing, co-registration to a standard template; MNI152). First-level analysis used a General Linear Model (GLM) approach with the four active conditions modelled as separate (event-series) regressors and the null trials implicitly modelled as the baseline. Also included were the first temporal derivatives of each time-series and an extended set of (24) head-motion parameters as regressors of no interest. Group level analyses computed a simple mean across all subjects within each scan session separately using FSL's FLAME-1 model and a statistical threshold of $Z = 3.1$, $p < 0.05$ (cluster-corrected). Contrasts were defined to isolate the response to each trial type relative to the null trials (baseline sections of the time-series). An additional analysis used mid-level fixed effects models to average each subject's responses across the two scan sessions, and then higher-level group analyses (with the same model and thresholds described above) to combine these data across subjects. These were conducted in order to visualize the overall effect of the tasks across subjects, and the two scans. Two separate sets of analyses were conducted, for data from the two task variants.

Additional analyses used Intra-Class Correlation (ICC; *Shrout & Fleiss, 1979*) coefficients to assess the reliability of responses across the two scanning sessions. This was performed in two ways; using an ROI-based approach, and by generating statistical maps of ICC values in a voxel-wise manner. For the ROI analysis, five regions were defined based on expected locations of brain activation in the tasks: primary auditory cortex in the superior temporal lobe (bilateral; auditory trials), primary visual cortex in the calcarine sulcus (bilateral; visual trials), left-hemisphere motor cortex (motor trials), the Frontal Eye Fields (FEF; bilateral; eye-movement trials), and the Dorso-Lateral Pre-Frontal Cortex (DLPFC; bilateral; working memory trials). ROIs were defined as 5 mm-radius spheres, and positioning coordinates were determined using guidance from relevant meta-analytic terms on Neurosynth (http://neurosynth.org/). The ROI definition was therefore performed completely independently from the main experimental data. Activation amplitude data was extracted from these ROIs for all subjects/scans and ICC(3,1) statistics were calculated using SPSS (IBM Corp; Armonk, NY).

The ICC statistical maps were produced using custom Python code and produced voxelwise images of ICC(3,1) statistics. For the purposes of thresholding the results, the ICC values were then transformed into standardized values ($Z$ scores) using the method of *Fisher (1915)*. These images were then thresholded using the same statistical criterion used for the group level BOLD activation analyses; $Z > 3.1$, $p < 0.05$ (cluster-corrected for multiple comparisons). These thresholded images were then used to mask the original ICC voxelwise images, to finally produce a robustly thresholded image, which also retains the original, more intuitive, ICC values. Several summary measures were extracted from these images. Firstly, the median ICC values in the thresholded images were calculated for each contrast/task condition. Secondly an analysis was performed in order to assess the relationship between the strength of activation, and the reliability of activation. Following the example of *Caceres et al. (2009)* median ICC values were calculated using images masks defined by different thresholds on the activation data. The activation data were thresholded at progressively higher $Z$ values (2.3, 3.1, 3.7, and 4.3; equivalent to $p = 0.01$, $p = 0.001$,

$p = 0.0001$, and $p = 0.00001$, respectively) and these image masks were used to calculate the median ICC scores using the unthresholded ICC images. In order to further examine the spatial relationship between the BOLD activation maps, and the reliability coefficient maps measures of the overlap (percentage of above-threshold voxels in the ICC maps, that were also above-threshold in the activation maps) were also calculated.

## RESULTS

### Behavioural performance

Subjects' behaviour was recorded and analysed to verify compliance with the task demands. An average accuracy rate of 93% was achieved within the working memory task. 94% and 97% accuracy was achieved for the motor task within the eye movement variant and working memory variant, respectively. All subjects performed the tasks satisfactorily.

### Group-level task activation

All tasks performed as expected and produced robust patterns of brain activity in regions previously shown to be activated by similar tasks. Performance of auditory, visual, and motor components of the tasks was consistent across both task variants (see Figs. 1A and 2A). Auditory trials produced strong bilateral activation within the superior temporal regions, consistent with primary auditory cortex (*Robson, Dorosz & Gore, 1998*). Visual trials produced activity in posterior calcarine sulcus and the occipital pole (primary visual cortex), and in the lateral visual region V5/MT+ (*Smith et al., 2006*; *Wall et al., 2008*). Motor trials produced activity in the left-hemisphere post-central sulcus, consistent with the known location of the hand representation in primary motor cortex (*Lotze et al., 2000*).

   In the eye-movement variant of the experiment, the eye-movement task produced activation in the Frontal Eye Fields (FEF), alongside activity within V5/MT+, the anterior portion of the calcarine sulcus/primary visual regions, and the intraparietal sulcus (see Fig. 1). This is generally consistent with previous reports of brain activity associated with eye-movement tasks. In the working-memory variant of the experiment, the working memory trials produced a highly robust activation pattern corresponding closely to that shown in conventional working memory tasks, such as the N-back (*Owen et al., 2005*). These regions included bilateral DLPFC, intraparietal sulcus, superior parietal lobule, dorsal anterior cingulate and the temporo-parietal junction (see Fig. 2).

   Parameter estimate data were extracted from each contrast using a set of five ROIs: primary auditory cortex (auditory trials), frontal eye-fields (eye-movement trials), left-hemisphere primary motor cortex (motor trials), primary visual cortex (visual trials), and dorsolateral-prefrontal cortex (working memory trials). These data are plotted for each condition and scan session in Fig. 3. Statistical analysis of these data used paired $t$-tests to compare data from each contrast across the two scanning sessions, and a Bonferroni-corrected alpha value of $p < 0.00625$ (corrected for 8 comparisons). None of the comparisons showed significant results.

### Reliability analyses

To assess voxel level reliability, intra-class correlation (ICC(3,1)) maps were created for each task (Figs. 1 and 2; right columns). These show a spatial distribution similar to the

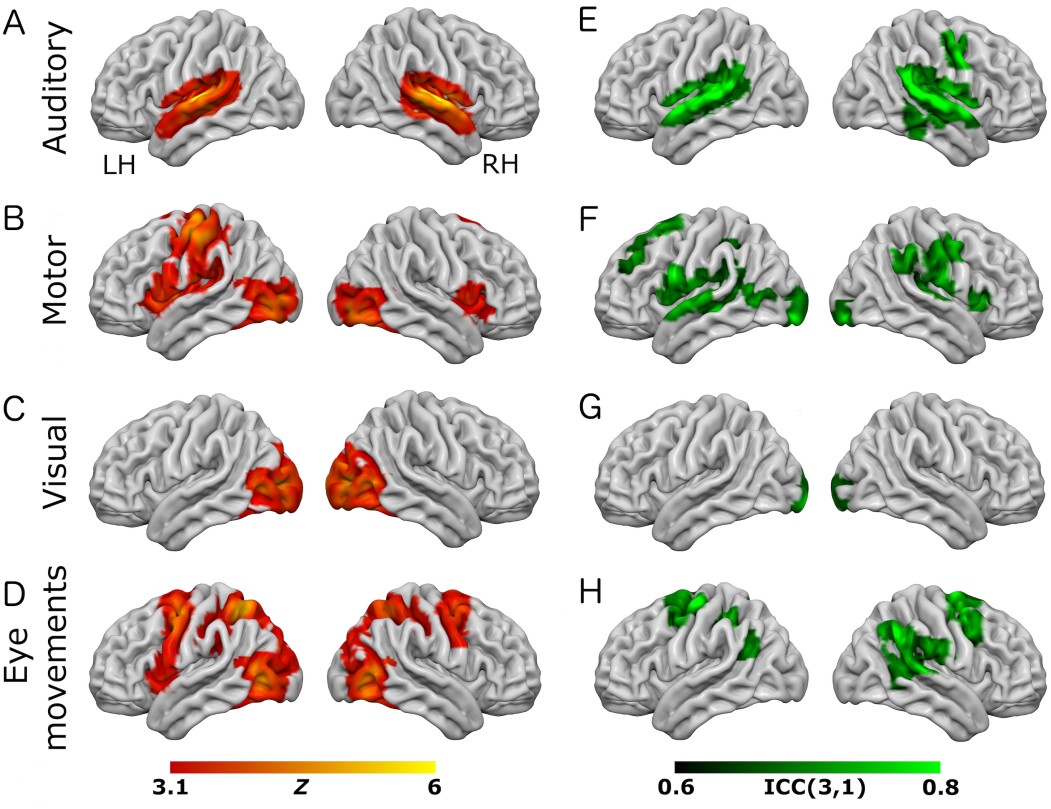

**Figure 1** **Results from the eye-movement variant of the task paradigm.** Results of group-level analyses represented on a cortical surface rendering of a standard anatomical image (MNI152). (A–D) Active brain regions for each contrast (mean of both scanning sessions) with functional maps thresholded at $Z > 3.1$, $p < 0.05$ (cluster-corrected). (E–H) Results of the reliability analysis comparing session 1 to session 2; Intra-class correlation (3,1) maps, masked with a Z-transformed, thresholded ($Z > 3.1$, $p < 0.05$; cluster-corrected) version in order to produce a robustly-thresholded image, while retaining the original ICC values (see methods for full details). (A, D) auditory trials; (B, E) motor trials; (C, F) visual trials, (D, H) eye-movement trials. See Tables S1 and S3 for cluster coordinates for all the statistical maps in this figure.

activation maps, with peak reliability estimates generally corresponding to the location of peak task-related activation. Reliability estimates in the working-memory variant of the task were generally higher and more widespread than in the eye-movement variant. The median ICC value in the thresholded maps is greater than 0.7 in all cases (see Tables 1 and 2 below). For additional visualizations of the spatial correspondence between the activation maps and the ICC results, see Figs. S1 and S2.

In the ROI analysis, one task condition (working memory) had a value >0.75 which is classed as 'excellent' under *Cicchetti*'s (*1994*) scheme for interpretation of ICC results. Three ROIs/conditions featured ICC values of 0.6 or above, which is classed as 'good'. A further three ROIs had values in the range 0.4–0.59 which is classed as 'fair' reliability. Only one (visual trials, in the eye-movement variant) was <0.4, and thus classed as 'poor'. The working-memory variant generally showed more consistent and higher reliability values than the eye-movement variant, in this analysis.

PeerJ

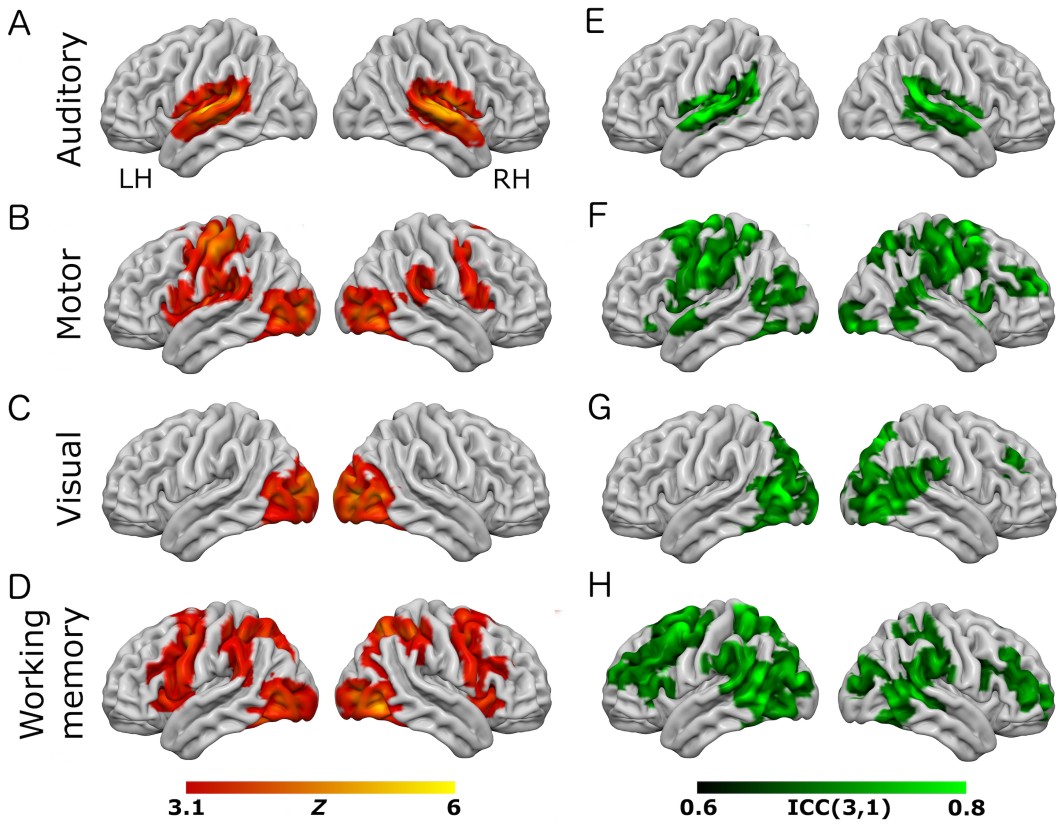

**Figure 2** **Results from the working-memory variant of the task paradigm.** Results of group-level analyses represented on a cortical surface rendering of a standard anatomical image (MNI152). (A–D) Active brain regions for each contrast (mean of both scanning sessions) with functional maps thresholded at $Z > 3.1$, $p < 0.05$ (cluster-corrected). (E–H) Results of the reliability analysis comparing session 1 to session 2; Intra-class correlation (3,1) maps, masked with a Z-transformed, thresholded ($Z > 3.1$, $p < 0.05$; cluster-corrected) version in order to produce a robustly-thresholded image, while retaining the original ICC values (see methods for full details). (A, D) auditory trials; (B, E) motor trials; (C, F) visual trials, (D, H) working memory trials. See Tables S2 and S4 for cluster coordinates for all the statistical maps in this figure.

In the analysis of ICC values at different activation thresholds, the activation data were progressively thresholded at higher levels, and these image masks used to produce median ICC values from the (unthresholded) ICC maps. Reliability measures from this procedure (see Tables 1 and 2) were generally poor, with only one task condition (auditory trials in the eye-movement variant) >0.6 (i.e., 'good' reliability) and many other values <0.4 (i.e., 'poor' reliability).

Unthresholded statistical maps resulting from all the group-level analyses (brain activation maps, and the voxel-wise ICC maps) are available to view at: https://neurovault.org/collections/3264/. All raw data from this study is also available at: https://openneuro.org/datasets/ds001344/versions/00001.

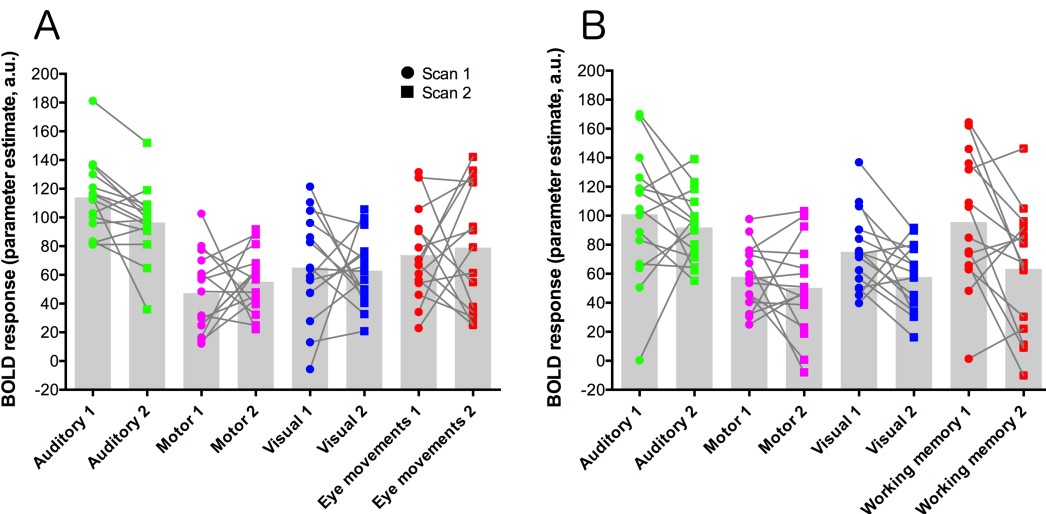

**Figure 3 ROI data for each task condition within the two task variants (A = eye-movement variant, B = working-memory variant).** Units are parameter estimates resulting from each of the four contrasts in each GLM analysis, relative to baseline (null trials) and are arbitrary units. ROIs are primary auditory cortex (auditory trials; green), left-hemisphere primary motor cortex (motor trials; magenta), primary visual cortex (visual trials; blue), the frontal eye-fields (eye-movement trials; red; (A), and dorsolateral-prefrontal cortex (working memory trials; red; (B). Grey histogram bars depict the mean of each condition. See Fig. S3 for images of the ROIs.

**Table 1 ICC(3,1) values for the eye-movement variant of the task.** The first column shows the median ICC values in the (thresholded) ICC(3,1) maps, for each task condition. The second column shows ICC(3,1) calculated from BOLD amplitude data in pre-defined ROIs. The next four columns show median ICC(3,1) values from the unthresholded ICC maps, within image masks produced by thresholding the task-activation maps at progressively more conservative levels ($Z = 2.3$, to $Z = 4.3$). The final column shows the percentage of voxels in the (thresholded) ICC(3,1) maps that are also above-threshold in the task activation maps.

| Task condition | Median of thresholded ICC(3,1) maps | ICC(3,1) values (in ROIs) | Median of ICC values in image masks produced by thresholding activation maps | | | | Overlap between ICC and activation maps (%) |
|---|---|---|---|---|---|---|---|
| | | | $Z = 2.3$ | $Z = 3.1$ | $Z = 3.7$ | $Z = 4.3$ | |
| Auditory | 0.77 | 0.59 | 0.61 | 0.66 | 0.68 | 0.67 | 66.0 |
| Visual | 0.7 | 0.17 | 0.14 | 0.24 | 0.27 | 0.26 | 55.5 |
| Motor | 0.71 | 0.75 | 0.23 | 0.22 | 0.18 | 0.12 | 7.2 |
| Eye-movement | 0.71 | 0.58 | 0.21 | 0.22 | 0.21 | 0.15 | 17.6 |

## DISCUSSION

We have developed and successfully validated two variants of a novel fMRI control task and demonstrated that they show good test-retest reliability. These tasks are short (five minutes duration), relatively simple for both the experimenter and subject (they require only standard audio-visual presentation equipment and a one-button response box), are highly robust in terms of the amplitude of brain activation produced, and show strong reliability features across two sessions. Each variant also produces a number of useful

**Table 2 ICC(3,1) values for the working memory variant of the task.** The first column shows the median ICC values in the (thresholded) ICC(3,1) maps, for each task condition. The second column shows ICC(3,1) calculated from BOLD amplitude data in pre-defined ROIs. The next four columns show median ICC(3,1) values from the unthresholded ICC maps, within image masks produced by thresholding the task-activation maps at progressively more conservative levels ($Z = 2.3$, to $Z = 4.3$). The final column shows the percentage of voxels in the (thresholded) ICC(3,1) maps that are also above-threshold in the task activation maps.

| Task condition | Median of thresholded ICC(3,1) maps | ICC(3,1) values (in ROIs) | Median of ICC values in image masks produced by thresholding activation maps | | | | Overlap between ICC and activation maps (%) |
|---|---|---|---|---|---|---|---|
| | | | $Z = 2.3$ | $Z = 3.1$ | $Z = 3.7$ | $Z = 4.3$ | |
| Auditory | 0.73 | 0.45 | 0.43 | 0.45 | 0.46 | 0.44 | 62.9 |
| Visual | 0.73 | 0.65 | 0.42 | 0.43 | 0.41 | 0.38 | 30.4 |
| Motor | 0.71 | 0.6 | 0.3 | 0.28 | 0.24 | 0.22 | 7.1 |
| Working memory | 0.72 | 0.76 | 0.4 | 0.39 | 0.35 | 0.23 | 9.5 |

readouts (visual, auditory, motor, cognitive/eye-movements) in a wide spatial distribution across the brain.

Both task variants performed similarly for visual, auditory, and motor trials, with robust activity seen in primary visual, auditory, and motor cortex respectively, and little 'off-target' activation evident. The eye-movement task also produced a characteristic pattern of brain activity similar to that seen in previous eye-movement studies (e.g., *Berman et al., 1999*). The working memory task, though only requiring a very brief (two-second) retention interval, produced a highly similar pattern of activity to that seen in more standard working memory tasks such as the N-back task (*Owen et al., 2005*).

Importantly, reliability of the tasks was also assessed, and found to be generally high. Reliability assessment using ICC (or other measures) is still relatively uncommon for fMRI experiments, but is an important step in validating task paradigms (*Caceres et al., 2009*). The ICC measures obtained here compare favourably with previous reports using auditory and working memory tasks (*Caceres et al., 2009*), a cognitive-emotive test battery (*Plichta et al., 2012*), and a reward task (*Fliessbach et al., 2010*). However, some task conditions were seen to be more reliable than others. In particular, reliability in the working-memory variant of the experiment was generally higher than in the eye-movement variant. One possible explanation for this difference may be due to the much more cognitively demanding features of the working-memory variant, which led to a higher level of attention and engagement to all the task conditions in that variant.

Previous work by *Caceres et al. (2009)* showed that the median ICC value tended to increase in regions defined by progressively higher activation thresholds; suggesting an association between higher BOLD activation, and higher reliability. This seems not to be the case in the current data, with profiles of ICC values remaining flat at different thresholds (see Tables 1 and 2, and Fig. S7) with relatively poor (though broadly comparable to *Caceres et al., 2009*) ICC coefficients, when derived in this way. Quantification of the spatial concordance between the activation and ICC maps showed that the sensory (auditory and visual) conditions share a much higher proportion of voxels than the other

(motor, cognitive) conditions, with rates of <10% for the latter in some cases. However, the median ICC values in these conditions are still high (>0.7). This suggests that there may be a distinction between brain regions which are robustly activated by a task, and brain regions which are *reliably* modulated by a task across two sessions, and that the extent of this dissociation also depends on the task/conditions used. The task conditions that produce activation in relatively focused and discrete brain regions (i.e., primary auditory and visual cortices) show higher spatial concordance than those tasks which activate much broader brain networks. As noted above, there is a paucity of studies that have explicitly assessed the test-retest reliability of fMRI, so much more work with different task paradigms and approaches will be required to further investigate these findings.

The high reliability, short duration, and ease of use of these tasks make them ideal for inclusion as control tasks in pharmacological-MRI studies, as suggested by *Iannetti & Wise (2007)*, and *Bourke & Wall (2015)*. Inclusion of tasks which are (hypothetically) unaffected by the drug helps rule out alternative explanations related to systemic drug effects (on blood pressure, heart-rate, etc.), or neuro-vascular coupling, which can both theoretically modulate the BOLD response. One previous study investigating modulation of amygdala responses by citalopram (*Murphy et al., 2009*) used a simple checker-board visual control task. Use of a single control task where activation is restricted to the occipital lobe is sub-optimal as the drug may potentially still produce non-neural effects in other brain regions. A recent study on the brain effects of the sex hormone kisspeptin (*Comninos et al., 2017*) used a control task with a number of readouts in different brain regions (based on *Pinel, Thirion & Meriaux, 2007*). This task was complex, with ten individual stimulus conditions, different response options, and contained high-level cognitive stimuli (performing mental arithmetic, reading sentences on the screen, and listening to recorded voices) which included culture- and language-specific features. This complexity and the use of language-specific stimuli limit the broad applicability of this task.

Use of control tasks in this manner can bolster support for a particular interpretation of a study's outcomes; however, they may never be conclusive. A successful outcome of these tasks (when used in the context of a pharmacological study) would be a *lack* of effect of the drug treatment, and strong inference from a null effect is an issue for standard statistical approaches based on null-hypothesis significance testing (*Neyman & Pearson, 1933*). Ideally, the experiment would be well-powered to detect such an effect, however recent meta-analytic work has demonstrated that low experimental power is an endemic problem in neuroscience research (*Button et al., 2013*), although see (*Nord et al., 2017*), for a somewhat more optimistic assessment). Use of Bayesian statistics is an alternative approach that may mitigate this issue and can provide a coherent method for determining whether a null result is 'genuine' or simply resulting from insensitivity of the data (*Dienes, 2014*; *Dienes, 2016*).

The tasks evaluated here represent a good compromise between ease of use, wide applicability, a short duration, reliable results, and the desirability of providing a number of readouts in spatially diverse brain regions. While the working memory variant appears to be somewhat more robust, more reliable, and produces a wider pattern of brain activity, it is also more cognitively demanding and has significantly more complex instructions.

This may make it less suitable for any patient group with significant cognitive impairments, who may struggle with a fast, demanding task. The eye-movement variant may therefore be more suitable for these groups. Additionally, the eye-movement variant may also be more suitable where the drug under investigation is hypothesized to have an effect on cognition. In this case, the working-memory variant may be inappropriate as a control task, as it strongly engages well-known cognitive brain regions. Either variant would also be suitable for use in a number of other situations where a short, reliable fMRI task that yields a number of readouts is required, for example in systematic testing of fMRI acquisition sequence parameters (as in *Demetriou et al., 2018*). Standardization of task paradigms, data acquisition practices, and analysis procedures in fMRI is currently at a nascent stage. Encouraging progress is being made on some aspects (e.g., *Esteban et al., 2018*) however the huge variety of experimental designs and analysis strategies in current use still impedes direct comparison of studies. Open-source tasks that have undergone some kind of formal evaluation or validation (such as those presented here) could therefore be a useful tool in future efforts at standardization. We have evaluated two variants of a novel task paradigm, suitable for use as a control task in pharmacological fMRI studies, or for any use where a general-purpose battery of basic tasks/stimuli is required. The tasks produce robust brain activation and have strongly favourable reliability features. The tasks are programmed in an open-source language and experimental presentation application (Python/PsychoPy), and we have therefore made the stimulus code freely available on Figshare at https://figshare.com/articles/fMRI_control_task_zip/5162065 (DOI: 10.6084/m9.figshare.5162065; short-link: goo.gl/DAqn4V), and on GitHub at: https://github.com/mattwall1103/fMRI-Control-Task (short-link: https://goo.gl/PcRurT). We encourage any interested researchers to download the programs and use them in their research.

## ACKNOWLEDGEMENTS

We would like to thank the Imanova Center for Imaging Sciences (Hammersmith Hospital, London, UK, Now Invicro, London) for the scanner time required to complete the project, and general support throughout the investigation

### Funding

This work was supported by Invicro Ltd. The funders had no role in study design, data collection and analysis, decision to publish, or preparation of the manuscript.

### Grant Disclosures

The following grant information was disclosed by the authors:
Invicro Ltd.

### Competing Interests

MBW and LD are primarily employed by Invicro Ltd., a private company which performs contract research work for the pharmaceutical and biotechnology industries. JM is

employed by Perspectum Diagnostics Ltd., also a private company, specialising in liver diagnostic imaging. JLH has no competing interests to disclose.

## Author Contributions

- Jessica-Lily Harvey conceived and designed the experiments, performed the experiments, analyzed the data, prepared figures and/or tables, authored or reviewed drafts of the paper, approved the final draft.
- Lysia Demetriou conceived and designed the experiments, performed the experiments, analyzed the data, authored or reviewed drafts of the paper, approved the final draft.
- John McGonigle analyzed the data, contributed reagents/materials/analysis tools, authored or reviewed drafts of the paper, approved the final draft.
- Matthew B. Wall conceived and designed the experiments, performed the experiments, prepared figures and/or tables, authored or reviewed drafts of the paper, approved the final draft.

## Human Ethics

The following information was supplied relating to ethical approvals (i.e., approving body and any reference numbers):

All scans were performed under local institution-approved guidelines for fMRI scans on healthy subjects to ensure adherence to ICH-GCP standards.

## Data Availability

Unthresholded statistical maps resulting from all the group-level analyses (brain activation maps, and the voxel-wise ICC maps) are available to view at: https://neurovault.org/collections/3264/. All raw data from this study is also available at: https://openneuro.org/datasets/ds001344/versions/00001.

Code for running the stimulus programs used in the experiments has been made available on FigShare at Wall, Matthew (2017): fMRI_control_task.zip. figshare. Code. https://doi.org/10.6084/m9.figshare.5162065.v1.

It is also available on GitHub: https://github.com/mattwall1103/fMRI-Control-Task.

## Supplemental Information

Supplemental information for this article can be found online at http://dx.doi.org/10.7717/peerj.5540#supplemental-information.

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
