# Peer review of "A short, robust brain activation control task optimised for pharmacological fMRI studies"

_PeerJ, doi:10.7717/peerj.5540_

## Round 0.1 · original submission · Minor Revisions

The expert reviewers provide clear comments. Please address each comment in your reply. In addition, I provide my own comments here.

I believe this is good work, and it is well suited to PeerJ. The authors have taken pains to be open with their work so that it can reach a wide audience.

I am skeptical about the focus on phMRI. Specifically, this whole project is an attempt to rescue BOLD-sensitive sequences from their basic deficit of being non-quantitative and useless for comparisons over more than a minute or two, comparisons that are necessary for phMRI with most drugs. (Our study https://peerj.com/articles/687/ highlights this concern.) Below are notes I wrote as I read through the manuscript. Addressing these thoughts in the ms. text, if possible, may strengthen the conclusions about phMRI. Otherwise, your study remains valuable but you could temper the enthusiasm about phMRI applications. Thank you and my comments follow.

ABSTRACT
* I think that developing and showing reliability of a suite of fMRI tasks that activate wide swaths of the brain is important and potentially very useful. Nevertheless, I question the conclusion that that accomplishment is relevant to phMRI. I would sell it differently.
* "the BOLD response is an indirect measure of neural activity, and as such is vulnerable to confounding effects of pharmacological probes." -- I'm not sure why indirect has anything to do with pharmacological. I guess you are saying, "... indirect ..., depending on intact coupling of activity to metabolism and then to blood flow, and as such ..."
* I'm not sure how these tasks actually control for pharmacological effects on the brain. For task fMRI, if the drug may have direct effects on brain activity, those directs are almost certainly regionally selective, and the activity in some other part of the brain is mostly irrelevant. The root problem is the choice of BOLD, which [in its more typical implementations] is quantitatively meaningless over periods of time more than a few minutes. "Either of the two task variants would be suitable for use as a control task in future pharmacological fMRI studies or for any situation where a short, reliable, basic task paradigm is required." -- the latter makes more sense to me.
* Why would neurovascular coupling effects of a drug be thought to differ in different brain regions? The vasculature is likely homogenous. Are the authors aware of a drug with such effects?

INTRO
* blood pressure is irrelevant, though it has occasionally been used as a poor substitute for rCBF. Heart rate is relevant only when a frequency of 60-100/min. could interfere in the statistical analysis with task paradigms that involve a similar frequency. Global effects on CBF can be controlled easily in analysis if measured, and if not measured, I fail to see how these tasks can control for them.
* Caffeine is not "selective" in the usual sense as it is an antagonist at both A1 and A2 receptors.

RESULTS
* overlap of thresholded statistical images is less compelling than direct comparison of BOLD amplitude similarity (because the former can change with changes in variance as well as changes in mean)

DISCUSSION
"Inclusion of tasks which are (hypothetically) unaffected by the drug helps rule out alternative explanations related to systemic drug effects (on blood pressure, heart-rate, etc.), effects on local vasculature, or neuro-vascular coupling; all of which can theoretically modulate the BOLD response." -- same question as above; you could rule out an effect by which a drug minimized BOLD changes in general, but how can you rule out focal effects of the drug, or how would you notice global effects on baseline blood flow?

= = = MINOR changes = = =
* "discreet" in several places in the text -> discrete
* "activation in a subscribed region of the brain" -> circumscribed, not subscribed
* "indirect effects on neurovascular coupling" -> direct (right?)

·

Basic reporting

See general comments

Experimental design

See general comments

Validity of the findings

See general comments

Additional comments

"A short, robust brain activation control task optimised for pharmacological fMRI studies" by Harvey et al. proposes a new standardized task fMRI paradigm optimized for short duration and ease of use. It has a potential for broad adoption in clinical studies.

Comments:
- Line 45: not clear what is the difference between “reliability coefficients for the tasks” and “Voxel-wise reliability metrics”. Maybe more intuitive labels could be used (ROI vs voxel)?

- I really enjoyed the outline of properties of a good pharmacological fMRI task. Maybe it would be good to highlight it in a box or a table. It makes a great guideline for future developments in this area.

- It might be beneficial to accompany the manuscript with video recordings of the visual and auditory stimuli of an example run for the two tasks. Such videos could be a great way to showcase the task to prospective users.

- It is not very clear how the instructions were delivered and if they are standardized. I could not find the instructions in the accompanied code. If this task is supposed to be broadly used in a clinical setting instructions should be explained better or automated.

- The manuscript would benefit from a figure describing the experimental paradigm.

- Why is the buffer time (10s) at the end rather than the beginning? Putting it at the beginning would help with potential non-steady state effects often found in EPI sequences.

- Line 175: missing comma after “31ms”

- Line 182: please specify which template was used

- Line 190: missing citation for ICC (TODO)

- Line 219: referencing figures is usually done with capitalized form (“Figures” vs. “figures”)

- Line 187: It seems that session effects were not modeled explicitly. It might be worth considering a model that removes session mean?

- Figures 1 and 2: I noticed that effects were tested and reported only in one direction (positive). This made me thinking – How would the negative effect look like? It would probably be task-negative network/default mode network. It might be worth looking at “any task vs null” contrast to see if this short protocol can also be used to reliably map that network.

- Figure 3: please consider drawing lines linking corresponding sessions 1 and 2 for each participant. You might also want to reconsider using yellow on white – it has poor contrast. (example: https://www.nature.com/articles/sdata201454/figures/3)

- Big kudos for sharing statistical maps on NeuroVault

- Line 364: It's great that authors share the code, but I don't think figshare is the best place to do it. Uploading the code to GitHub and using zendo for long-term preservation would allow users of this paradigm to submit improvements and bugfixes.

- Source code should specify all of the dependencies with their versions. This could be easily done with "pip list" or "conda list"

- Finally, I strongly encourage the authors to share the raw data from this study. This should be relatively easy on a dedicated platform such as https://OpenNeuro.org

·

Basic reporting

Very well written and super clear manuscript.

Experimental design

The authors present a novel paradigm with a specific goal described of use in pharmacological imaging studies. In relation to previous studies that have used single tasks the logic is clear and the aims are good. As I understand the authors present the relatively straightforward idea that a short, multi component task and this leads to reliable activations across a variety of brain regions. By using this is a drug study then the LACK of effect on this task can be taken that the drug does not have non-specific vascular effects. This is an improvement on a small number of other studies (one is given as an example) that only tested a single modality in the control task.

There are two issues not address by the authors in presenting this framework. The first is that the fMRI tasks of interest in any study already include control conditions and therefore are intrinsically controlling for ‘non-specific’ effects of drugs. The authors need to articulate how an additional control task assists, for example drawing on the issues of task design and ‘controls of controls.’ The second issue is that the logic requires the control paradigm to show NO effect of the drug and therefore is subject to problems in statistical inference. What is the statistical framework in which the authors see this task being used? Bayesian, interaction effects, something else? A comment on its implementation would be welcome.

The description of the methods could be improved. Was there a training session? Was the paradigm sequential blocks of the different modalities or were they intermixed? If blocked, were they randonmised? Was the data analysed as blocks or event-series?

Validity of the findings

The data processing and analysis are otherwise clear. There are current changes in practice suggested by the literature that are not incorporated. Please justify why not, or instead consider updating the analysis. First, the movement appears to be modelled using only the six parameters (but this is not explicitly stated). Using up to 24 parameters is proving to be better in terms of reducing noise and preserving signal, as are other methods such as ICA-FIX/AROMA. Second, a cluster-forming threshold of z=3.1 is used. The well-known paper by Eklund et al. (www.pnas.org/cgi/doi/10.1073/pnas.1602413113) demonstrated that in FSL this gives elevated false positive rates of clusters. Please consider using different cluster-forming thresholds. Given the reliability output, using more than one threshold in FLAME could provide useful information.

For the ICC and Z maps the authors claim good overlap, but it is not clear what this is based on. Visual inspection? Proportions? Numerically indexing the overlap would be important as it is difficult to tell from the images alone, which contain substantial red and green (i.e. no-overlap) voxels. Does it matter that they overlap?

Additional comments

The manuscript would benefit from a table of coordinates for the peaks in activation and ICC clusters, including the extents.

The lower reliability in the eye-movement task is discussed in lines 324-329. Is it possible that another reason is the inclusion of error trials? It is not said what was done with these even though they were only 6 % of trials on average.

The availability of the task and data is a strength as good reliability data sets are few and far between.

---

## Round 0.2 · accepted · Accept

I am not persuaded about the major questions I raised about the conclusions with regard to pharmacological challenges, but your methods and arguments will likely persuade others, and in any case they will be useful in other applications as well.

I would strongly urge you to make the reviews and your response letter public, as I think the discussion (and your task-induced BOLD decreases data) will benefit readers.

You asked if it is OK to add a text box. Yes, no objection on my part. You can submit it as a table, or the PeerJ staff can correct me on how best to do this.

#